# Mechanical Behavior of Austenitic Steel under Multi-Axial Cyclic Loading

**DOI:** 10.3390/ma16041367

**Published:** 2023-02-06

**Authors:** Abhishek Biswas, Dzhem Kurtulan, Timothy Ngeru, Abril Azócar Guzmán, Stefanie Hanke, Alexander Hartmaier

**Affiliations:** 1Interdisciplinary Centre for Advanced Materials Simulation (ICAMS), Ruhr-Universität Bochum, Universitätstraße 150, 44801 Bochum, Germany; 2Chair of Materials Science and Engineering, Universität Duisburg-Essen, 47057 Duisburg, Germany

**Keywords:** multi-axial fatigue, crystal plasticity, micromechanical modeling, austenitic steel

## Abstract

Low-nickel austenitic steel is subjected to high-pressure torsion fatigue (HPTF) loading, where a constant axial compression is overlaid with a cyclic torsion. The focus of this work lies on investigating whether isotropic J2 plasticity or crystal plasticity can describe the mechanical behavior during HPTF loading, particularly focusing on the axial creep deformation seen in the experiment. The results indicate that a J2 plasticity model with an associated flow rule fails to describe the axial creep behavior. In contrast, a micromechanical model based on an empirical crystal plasticity law with kinematic hardening described by the Ohno–Wang rule can match the HPTF experiments quite accurately. Hence, our results confirm the versatility of crystal plasticity in combination with microstructural models to describe the mechanical behavior of materials under reversing multiaxial loading situations.

## 1. Introduction

Austenitic stainless steels, like 316L and Rex 734, exhibit remarkable corrosion resistance and formability, which makes them suitable for medical implants. However, the leaching of alloying metals like chromium, cobalt, and nickel ion in the human body environment can cause a strong allergic response and is even found to be carcinogenic in rats [1,2,3]. This led to the development of a low nickel and high nitrogen austenitic stainless steel 1.3808 with the commercial name P558, which exhibits remarkable mechanical properties and good bio-compatibility [4,5,6,7] with targeted applications in medical implants. In view of the excellent properties of P558, many advanced manufacturing techniques like liquid-phase sintering [8] and metal powder injection molding [9] have been explored to manufacture complex components. However, the existing research of austenitic stainless steel mechanical properties is mainly focused on stainless steel grades, like 316L or 304L, and few publications report even very basic mechanical properties like yield stress, surface roughness, or hardness of P558 [9,10].

The understanding of material behavior under multiaxial mechanical loading is of great importance for the application of structural materials because components are oftentimes exposed to multiaxial stress states. In addition, the loading is typically time-dependent and reversing. Experimental investigations under such complex loading conditions are very laborious, such that our understanding of the mechanisms leading to plastic deformation and failure of materials under multiaxial reversing loading is quite insufficient. This is particularly important for a newly developed steel, like P558, and the present work aims at closing this gap in our understanding of the mechanical behavior of this new austenitic steel under complex mechanical loading.

To investigate the kinematic hardening behavior of P558 austenitic stainless steel under multiaxial reversing loading, the steel specimens are subjected to combined compression and cyclic torsional load. In preliminary investigations, it has been observed experimentally that, under combined compression and cyclic torsional loading, the specimens undergo plastic axial strain although the compressive load is well below the yield stress of the material. This time-dependent deformation under small stresses can be considered as room-temperature creep behavior, triggered by the multiaxial loading. This compressive creep occurs as soon as a critical angle for the cyclic torsion is exceeded, such that it remains unclear whether an equivalent stress of the applied stress tensor can be defined that describes this deformation mode with sufficient accuracy. Ngeru et al. [11] reported similar behavior in a high nitrogen stainless steel subjected to similar loading conditions. It is noted here, that this kind of creep deformation is purely mediated by dislocation motion, whereas diffusive or other thermally activated mechanisms do not play a role. The term creep is merely used here to describe time-dependent plastic deformation occurring at a constant stress below the yield stress of the material.

Numerical simulations of the mechanical behavior of metals under cyclic loading have been reported frequently in the literature, e.g., Cruzado et al. [12] performed crystal plasticity finite element model (CPFEM) simulations of a polycrystalline microstructure and showed a good agreement between simulation and experiments. In other work, the effects of monotonic-cyclic loading on the behavior of high-alloyed steel and pure copper were investigated and an axial stress-drop was reported for combined cyclic torsion and monotonic axial tensile loading [13].

To incorporate kinematic hardening in constitutive models, mathematical descriptions are required that can predict both the stress–strain hysteresis as well as the relaxation of the mean stress that occurs under cyclic loading [14]. During strain-controlled cyclic loading, the mean stress is defined as the average between the maximum and the minimum stress resulting from the cyclic material response. If the upper and the lower value of the applied strain are not equal with respect to their absolute value, i.e. for asymmetric loading with amplitude ratios different from R=−1, the mean stress typically takes a rather high positive or negative values at the beginning of the cyclic experiment, but then it usually relaxes towards zero. The importance of the correct description of kinematic hardening in constitutive models for multiaxial cyclic loading has been demonstrated for investigations on thin-walled tubular specimens under axial tension, and cyclic torsion [15]. To describe kinematic hardening in general, the empirical kinematic hardening models of Armstrong–Fredrick [16], Chaboche [17], and Ohno–Wang [18] are most widely used. Several works implying J2 plasticity [18,19] or crystal plasticity [14] have noted that the Armstrong–Fredrick kinematic hardening model failed to predict the mean stress relaxation that occurs under non-symmetric loading conditions, where the amplitude ratio R≠−1. Schäfer et al. [20] showed a comparison between different kinematic hardening models in CPFEM with loading cases having both symmetric loading R=−1 and non-symmetric loading R=0. The results indicate that the Ohno–Wang model is most adept at predicting cyclic mechanical properties. Sajjad et al. [21] used both, J2 as well as CPFEM, with the Chaboche kinematic hardening model for predicting the hysteresis loop and the mean stress relaxation under uniaxial cyclic loading. Based on the aforementioned publications, in this work, the Chaboche kinematic hardening model is used with J2 plasticity, and the Ohno–Wang kinematic hardening model is selected for the CPFEM simulations due to their ability to predict complex cyclic properties for different loading cases and *R*-values.

The outline of the present work is as follows: Section 2 describes the experimental setup for the EBSD and fatigue experiments. Section 3 explains the modeling strategy for the estimation of elastic stiffness constants, the micromechanical modeling based on experimental input, and the material models used to mimic the material behavior in the numerical model. Section 4 shows the comparison of the results from the numerical model and experiments and provides a discussion of the results and observations. Finally, in Section 5, the conclusions of this work are laid out. To support the reading of the mathematical equations, a list of variables/symbols is provided in the Nomenclature section.

## 2. Material Characterization

### 2.1. Specimen Preparation and Setup of EBSD

P558 is the commercial name for the austenitic stainless steel 1.3808 sold by voestalpine BÖHLER Edelstahl GmbH & Co KG (Kapfenberg, Austria). The raw material was delivered in the form of cylindrical bars from which the specimens were produced as specified below. The chemical analysis of the tested material is given in Table 1. Due to the absence of nickel in the alloy, a combination of excellent mechanical properties and high corrosion resistance is achieved, as well as good bio-compatibility. To bring the material into a well-defined state and to remove all cold-deformation from the sample manufacturing process, the material was solution annealed at 1150 ${}^{\circ }$C for 45 min. After that, tensile tests of the material resulted in an ultimate tensile strength of 937 MPa and a yield stress of 595 MPa, where the results of three standard tensile tests have been averaged.

For a microscopic analysis, specimens from bar material were cut both in longitudinal and cross-sectional directions according to Figure 1. Consequently, the material was embedded with conductive resin Technotherm 3000 (Kulzer GmbH, Hanau, Germany), ground up to 800 grit, and polished with silica on RotoPol-31 (Struers GmbH, Willich, Germany). The final preparation step for EBSD was vibration polishing of specimens at a frequency of 90 Hz for 3 h on Saphir Vibro (ATM Qness GmbH, Mammelzen, Germany).

A Field Emission Gun Scanning Electron Microscope (LEO Gemini 1530 from Carl Zeiss AG) was used to capture micrographs with the electron backscatter detector (EBSD) at a voltage of 20 kV and an aperture size of 120 μm. The working distance was 15 mm, and the spot size varied between 0.5 μm and 2 μm. The EBSD data sets were initially analyzed with the help of the analysis software OIM 6.2 from EDAX (Mahwah, NJ, USA), due to the large number of EBSD data sets, MATLAB scripting (with MTEX toolbox) was used for the post-processing of data sets.

### 2.2. Microstructure Analysis Using EBSD

In this section, we describe the microstructural features observed in the EBSD data sets, which provide the key inputs for the generation of the virtual microstructure used for CPFEM simulations. To properly characterize the microstructure across the bulk material, 23 EBSD measurements were performed at different locations of the longitudinal and cross-sectional cuts and for different magnifications, resulting in the characterization of ≈19,000 grains in the underlying microstructure.

Figure 2 shows the (Inverse pole figure) IPF-Z plot of one representative EBSD map and the contour plot of the pole figure of the orientation distribution function (ODF) estimated by combining 23 EBSD data sets. The ODF is estimated using a kernel density estimate [22].
(1)$$f\left(q\right)=\sum_{i=1}^{N_{g}}w_{i}\psi_{\Omega }\left(q_{i}\right)$$
where the function *f* is the ODF, *q* is the crystallographic orientation, $N_{g}$ is the number of grains, and $w_{i}=a_{i}/\sum a_{i}$ is the weighting factor for each grain estimated considering the area $a_{i}$ of the grain. The grains are estimated using a disorientation threshold of $5^{\circ }$, and to avoid experimental/estimation errors the grains smaller than 3μm are discarded. The grain size is represented using the equivalent diameter, which is the diameter of a circle with the same area ($=2\sqrt{\left(a_{i}/\pi \right)}$).

The microstructure exhibits mostly equiaxed grains containing annealing twins, which are formed due to the re-crystallization during the annealing process and which show a density that is proportional to the strain level achieved before the annealing process [23]. Usually, the annealing twins are of Σ3 type with both a symmetrical and asymmetrical grain boundary tilt [24].

### 2.3. Fatigue Test

Both uniaxial and high-pressure torsion fatigue (HPTF) experiments were conducted on cylindrical double-cone specimens, presented in Figure 3a,b. While uniaxial fatigue specimens have a shorter gauge length, the HPTF specimens possess a thicker gauge section to prevent the buckling of the specimens due to the multiaxial loading case. Before experiments, the gauge section of all specimens was ground with increasingly finer grades of sandpaper and polished with 1 μm diamond paste to remove turning marks. After polishing, the arithmetic mean roughness ($R_{a}$) of the specimen surface was 46±6 nm.

A Bionix 858 (MTS Corporation, Eden Prairie, MN, USA) servo-hydraulic universal testing machine was used for all fatigue tests. With the stated testing setup, the specimen can be loaded with 50 kN of maximum axial force and 300 Nm of maximum torsion moment. Uniaxial fatigue tests were performed at $20\pm 2{\;}^{\circ }$C with 40–60% relative humidity under total strain-controlled tension-compression with sinusoidal loading at R=−1. The frequency of the experiments was 2 Hz. All experiments were conducted until failure or run out at $2\times 10^{6}$ cycles.

The frequency of experiments was also controlled, limiting the temperature increase of specimens to 50 K for uniaxial tests (2.5 Hz for ϵ<1.5% and 0.5 Hz for ϵ=1.5%). During the HPTF tests, however, the temperature increase may reach values over 200 K and saturated after 400–500 cycles.

Multiaxial loading of the specimens was rotation angle controlled torsion ($0^{\circ }$ to 5–20${}^{\circ }$) with sinusoidal loading at R=0 or −1 and a frequency of 2.5 Hz, combined with a monotonous constant compressive stress 250 or 350 MPa. Figure 3 visualizes the special multiaxial loading of the specimen during experiments. The axial force necessary to apply the stated axial stress is related to the smallest cross-section of specimens, measured with a laser thickness gauge with a precision of 10 μm. The experiments are terminated with the observation of clearly visible cracks on the surface of the specimen. During the HPTF experiments, the specimens exhibit compressive axial strains under constant load below the yield stress, which is referred to as axial creep deformation in this work.

To have a proper database for the modeling of the material behavior under HPTF loading for different *R* values and strains, three different loading conditions were selected: ($R=-1,\;\theta =7.5^{\circ }$), ($R=0,\;\theta =10^{\circ }$), and ($R=0,\;\theta =15^{\circ }$) where θ is the amplitude of the cyclic torsion. Figure 4 shows the experimental results for the stress–strain hysteresis loops after the saturation of the mean stress and the axial creep for the selected HPTF loading cases. The equivalent true strain shown in Figure 4 is estimated using
(2)$$\epsilon^{\mathrm{true}}=\log\left(1+\epsilon^{\mathrm{eng}}\right),$$
where the engineering strain $\epsilon^{\mathrm{eng}}$ for torsion is defined as
(3)$$\epsilon^{\mathrm{eng}}=\theta \;\frac{r_{g}}{l_{g}},$$
with the radius $r_{g}$ and the length $l_{g}$ of the gauge section. This equivalent true strain, thus, corresponds to the strain reached at the surface of the gauge section. Similarly, the engineering shear stress $\tau_{eng}$ at the surface of the gauge section is defined in a linear approximation as
(4)$$\tau^{eng}=T\frac{r_{g}}{J},$$
where $J=0.5\pi r_{g}^{4}$ is the polar moment of inertia for the circular cross-section and *T* is the experimentally measured torque. The general expression for true stress is defined as
(5)$$\sigma^{true}=\sigma^{eng}\left(1+\epsilon^{eng}\right).$$

Table 2 summarizes the details of the fatigue experiments performed in this work, where θ denotes the amplitude of torsion cycles for HPTF tests and $E_{33}$ is the loading amplitude in the axial direction (*z*-axis) for uniaxial fatigue tests.

## 3. Micromechanical Model

In this section, the workflow of the micromechanical modeling is described. Due to the recent development of the P558 austenitic stainless steel, the stiffness constants essential for numerical modeling are unavailable in the published literature. Therefore, density functional theory (DFT) calculations are employed to estimate the elastic constants, as detailed in Appendix B. To incorporate the influence of various microstructural features such as inclusions, defects, etc., the ideal stiffness tensor estimated by the aforementioned DFT method using a pristine molecular structure is scaled by a scalar constant λ, such that $C_{\mathrm{eff}}=\lambda C_{\mathrm{DFT}}$). The parameter λ is calibrated to match the elastic part of the experimental hysteresis loop with corresponding FE simulations. $C_{\mathrm{eff}}$ is used to describe anisotropic elastic behavior in CPFEM and, in the case of the J2 model, isotropic elastic constants are determined by homogenizing the elastic stiffness tensor according to the crystallographic texture as quantified by f(q) [25]. Table 3 shows the ideal elastic stiffness parameters obtained with DFT calculations, homogenized Young’s modulus (*Y*) (without scaling), and the scaling factor λ. The values obtained from DFT simulations are comparable to the elastic modulus of 155–220 GPa reported for similar nickel-free stainless steels [26,27].

The microstructure characterization of the present material with EBSD reveals the presence of annealing twins. Castelluccio and McDowell [28] indicate that the annealing twins can influence the fatigue crack initiation. Furthermore, Pande et al. [29] demonstrate that size effects can occur due to annealing twins in the microstructure. However, this work focuses on cyclic property prediction rather than aiming at predicting fatigue failure. Therefore, for the sake of simplicity, annealing twins in the micromechanical model and application of gradient-based plasticity models to describe size effects are disregarded in the present work.

### 3.1. Representative Volume Element

Due to the high computational effort required for CPFEM simulations with kinematic hardening, a simple representative volume element (RVE) is selected in which each grain is represented by a cubic shape. To optimize the number of elements in the model, a parametric study of the RVE focusing on the number of elements used for the discretization of each grain is performed. This study indicates that an RVE with 512 grains and 8 elements per grain, see Figure 5, provides the optimum balance between mechanical property prediction and computational effort. The edge length of each cubic grain is taken as 0.05 mm, which is estimated as the mean equivalent grain diameter from the EBSD analysis, see Figure 2.

The required number of 512 discrete orientations for the grains constituting the RVE are systematically sampled from the EBSD data for a total of 19,000 grains, using the method in Biswas et al. [30]. The results of this sampling method are shown in Figure 5, and the quality of the sampling is judged using the error function $∥f\left(q\right)-\hat{f}{\left(q\right)∥}_{1}$, where ${∥\cdot ∥}_{1}$ is the $L_{1}$ norm of the difference of experimental ODF f(q) and the reduced ODF $\hat{f}\left(q\right)$ used for the RVE simulations. It is seen that sampling of 512 discrete orientations is achieved within an error margin of 10%. Since this work focuses on the macroscopic properties, the grain boundary disorientation distribution is neglected [31].

### 3.2. Material Model

#### 3.2.1. J2/Von Mises Plasticity

In this section, the J2 model with the isotropic hardening/softening and the Chaboche kinematic hardening model [17] are described briefly. For a detailed description please refer to ABAQUS documentation [32]. The yield function
(6)$$f\left(\sigma \right)=\sqrt{\frac{3}{2}\left(\sigma^{\mathrm{dev}}-\kappa^{\mathrm{dev}}\right):\left(\sigma^{\mathrm{dev}}-\kappa^{\mathrm{dev}}\right)}-\sigma^{m},$$
depends on the stress tensor σ and incorporates kinematic hardening via the back-stress κ. The scalar yield stress of the material is denoted as $\sigma^{m}$, and the superscript “dev” indicates that the corresponding deviatoric tensor is addressed. Plastic yielding sets in when the yield function vanishes, i.e., f(σ)=0 is used as the yield criterion. If the value of the yield function is larger than zero, a plastic strain increment is calculated that reduces the elastic strain and, thus, relaxes the stress to a value where the yield function is again zero. In this so-called return mapping algorithm, the plastic strain increments are calculated as
(7)$$d\epsilon_{\mathrm{pl}}=d\lambda \;\frac{\partial f}{\partial \sigma }=\frac{3}{2}d\lambda \;\frac{\sigma^{\mathrm{dev}}-\kappa^{\mathrm{dev}}}{\sqrt{\frac{3}{2}\left(\sigma^{\mathrm{dev}}-\kappa^{\mathrm{dev}}\right):\left(\sigma^{\mathrm{dev}}-\kappa^{\mathrm{dev}}\right)}}\;,$$
where the direction of the plastic strain is given by the gradient of the yield function with respect to the stress tensor and the magnitude of the plastic strain is quantified by the plastic strain multiplier dλ. It is noted here, that for this associated flow rule the direction of the plastic strain increment corresponds to the deviatoric stress tensor reduced by the back stress tensor. For details, the reader is referred to the basic literature on continuum plasticity, e.g. the textbook of de Borst et al. [33].

The isotropic hardening/softening is modeled using an exponential law as
(8)$$\sigma^{m}=\sigma^{m}+Q\left(1-e^{-b\epsilon_{eq}}\right),$$
where *Q* and *b* are material parameters that can be estimated by matching experimental results, and $\epsilon_{\mathrm{eq}}=\sqrt{\left(2/3\right)\epsilon_{\mathrm{pl}}:\epsilon_{\mathrm{pl}}}$ is the equivalent plastic strain. The evolution of κ is given by
(9)$${\dot{\kappa }}_{i}=C_{i}\;{\dot{\epsilon }}_{\mathrm{eq}}\;\frac{1}{\sigma^{m}}\left(\sigma -\kappa_{i}\right)-g_{i}{\dot{\epsilon }}_{\mathrm{eq}}\;\kappa_{i}\;,$$
where $C_{i}$, $g_{i}$ are the material parameters, and ${\dot{\epsilon }}_{\mathrm{eq}}$ is the equivalent plastic strain rate. The index i∈{1,2} addresses two different back-stresses terms considered here. The total back-stress is computed as $\kappa =\sum_{i}\kappa_{i}$.

#### 3.2.2. Crystal Plasticity

This section presents a brief description of the crystal plasticity model used in FE simulations of the mechanical behavior of the RVE. For the sake of brevity, we mainly focus on the modeling of kinematic hardening, please refer to [34] and references therein for a detailed description of the CPFEM method. The total deformation gradient is described using multiplicative decomposition $F=F^{e}F^{p}$, where $F^{e}$ is the elastic part representing a reversible lattice deformation and rotation, and $F^{p}$ is the plastic part representing an irreversible lattice deformation.

The plastic deformation is assumed to result from the motion of dislocations on crystallographic slip systems described by the slip direction $d^{\alpha }$ and slip plane normal $n^{\alpha }$ where $\alpha =1,...,N_{s}$ is the slip system index, and $N_{s}$ is the number of slip systems. The driving force for the motion of dislocations is given by the resolved shear stress ($\tau^{\alpha }$), which is calculated by mapping the second Piola-Kirchhoff stress tensor $S=\frac{C_{\mathrm{eff}}}{2}\left({F^{e}}^{T}F^{e}-I\right)$ on the slip system α. I is second rank unit tensor.

In this work, a phenomenological constitutive model is used, in which the kinematic hardening is modeled by including an additional back-stress $\tau_{b}^{\alpha }$ in the flow rule
(10)$${\dot{\gamma }}^{\alpha }={\dot{\gamma }}_{0}{\left|\frac{\tau^{\alpha }-\tau_{b}^{\alpha }}{\tau_{c}^{\alpha }}\right|}^{p_{1}}\mathrm{sign}\left(\tau^{\alpha }-\tau_{b}^{\alpha }\right)$$
where ${\dot{\gamma }}^{\alpha }$ is the shearing rate, $\tau_{c}^{\alpha }$ is the slip resistance for isotropic hardening. In this work, the Ohno–Wang model is used, which is adapted from the original formulation to the crystal plasticity model by defining the evolution of the back-stress as
(11)$${\dot{\tau }}_{b}^{\alpha }=\eta {\dot{\gamma }}^{\alpha }-\mu \frac{|\tau_{b}^{\alpha }|}{{\left(\eta /\mu \right)}^{m}}\tau_{b}^{\alpha }\left|{\dot{\gamma }}^{\alpha }\right|$$
following the work of McDowell [35]. Hennessey et al. [14] used two components for $\tau_{b}^{\alpha }$ using sets of {$\eta_{i}$, $\mu_{i}$, $m_{i}$ } with (i=1,2). For the sake of simplicity, in this work $\tau_{b}^{\alpha }$ is limited to a single component with only one set of {η, μ, *m*} as parameters. The isotropic hardening is given by the evolution of the slip resistance, described as
(12)$${\dot{\tau }}_{c}^{\alpha }=\sum_{\beta =1}^{N_{s}}h_{0}\xi_{\alpha \beta }{\left(1-\frac{\tau_{c}^{\beta }}{\tau_{c}^{f}}\right)}^{p_{2}}\left|{\dot{\gamma }}^{\beta }\right|$$
where $\tau_{c}^{\alpha }$ is the saturation slip resistance and $\xi_{\alpha \beta }$ is the cross-hardening matrix in which the diagonal elements representing the coplanar slip systems are set to 1.0, and off-diagonal elements representing the non-coplanar slip systems are set to 1.4.

## 4. Results and Discussion

In this section, the results obtained from numerical models are compared with experimental findings. Due to the multiaxial and heterogeneous stress and strain fields in the gauge section of the specimens, the material parameters cannot be assessed directly from the results of the HPTF test, but the constitutive parameters for J2 and CPFE model must be obtained using an inverse procedure instead, as shown in Appendix C. In the first step, a FE model of the gauge section of the test specimens is applied to characterize the material response under HPTF loading using a J2/von Mises plasticity model with Chaboche kinematic hardening and isotropic softening. In the second step, a representative volume element (RVE) is introduced that considers the mean grain size and crystallographic texture resulting from the EBSD analysis of the tested specimens, as described above. In FE simulations of this RVE, boundary conditions are applied that mimic the HPTF loading on a volume element close to the surface of the gauge section of the experimental specimens. The plastic deformation of each grain in the RVE is described by the CP model introduced above. It is noted here that in the present study, we do not attempt to model the grain shapes or even microstructural details as annealing twins, but we focus on the determination of the material parameters, which requires the use of a numerically efficient model. Furthermore, a parametric study of the CP kinematic hardening parameters is shown in Appendix A, emphasizing the importance of axial creep experiments for the calibration of parameters.

### 4.1. J2 Plasticity

The parameters for the J2 model are calibrated using data from uniaxial fatigue experiments. The parameters controlling the stress–strain hysteresis loop, i.e., $C_{1}$, $C_{2}$, $g_{1}$, and $g_{2}$, are obtained in an iterative process in which the difference between experimental and numerical hysteresis loops are minimized by systematically varying those material parameters. For this purpose, a representative hysteresis loop is chosen from the regime of load cycles in which the mean stress has already saturated. In contrast, the isotropic softening parameters, i.e., $\sigma^{m}$, *Q*, and *b*, are gained from the comparison of the maximum stress obtained during each load cycle between simulation and experiment. To obtain the hysteresis loop in the regime where the amplitude is constant, the experimental hysteresis loop from the 5000th cycle is used for calibration of the kinematic hardening parameters. The isotropic softening parameters are calibrated by simulating the initial 300 cycles.

Figure 6 shows the comparison of the saturated hysteresis, i.e., the hysteresis loop in the regime where the stress amplitude is constant, and the maximum stress obtained from simulation and experiments; the corresponding calibrated J2 material parameters are given in Table 4.

Keeping the J2 material parameters constant, HPTF simulations are performed with the same FE model, see Figure 7, in which the resulting axial creep strain $E_{33}$ obtained from the experiment and the HPTF simulations also are shown. It is seen that the simulations result in a similar axial creep behavior under different HPTF loading conditions, which stands in stark contrast to the experiment where the axial creep behavior is very sensitive to the torsional loading. Furthermore, the axial creep strains obtained from the FE simulations with material parameters fitted to uniaxial fatigue experiments grossly overestimate the experimentally found creep strains under HPTF loading, which points out the limitations of the J2 plasticity model. The reason for these deficits in the J2 model in predicting the axial creep behavior correctly can be attributed to the associated flow rule used here, see Equation (Equation 7), where the plastic strain increment is proportional to the deviatoric stress reduced by the back stress. Hence, as long as constant stress is applied along the axial direction, the plastic strain in this direction will not be sensitive to the amplitude of the cyclic torsional load, which is clearly seen in the simulation results for axial creep in Figure 7. Since this is a fundamental restriction, no attempt is made to test other modeling strategies for J2 plasticity, although in the literature, it has been demonstrated that an associated flow rule in combination with a three-surface hardening-recovery model for kinematic hardening could be successfully parameterized to describe the material behavior of thin-walled tubular specimens under monotonic-cyclic loading [15].

To understand the distributions of axial stress and axial plastic strain in the J2 model, these quantities are shown along a cut through the gauge section in Figure 8. The contour plots indicate that both, axial stress and plastic strain, are rather constant in each cross-section of the model. The axial stress, however, exhibits elevated compressive stresses close to the symmetry axis, where high hydrostatic stress components are observed.

### 4.2. Crystal Plasticity

Since the RVE with 512 grains is much smaller than the dimension of the experimental specimens, we cannot directly apply torsional loading on this RVE. Instead, we consider a small volume element close to the surface of the cylindrical FE model representing the gauge section of the experimental specimens, which has been used for the J2 plasticity calculations, see Figure 9. The distortions experienced by this volume element during the different HPTF loading cases are applied as boundary conditions to the CPFEM model to achieve a comparable loading situation on the length scale of the RVE. The major distortion occurs in the X–Z and Y–Z shear directions (tensor components $E_{13}$ and $E_{23}$, respectively). The RVE is subject to displacement boundary conditions resulting in X–Z shear, and a distributed force boundary condition in the Z direction, the magnitude of the force is adjusted such that the homogenized stress in the RVE matches the required axial pressure for the given HPTF load, and similarly, the shear displacement is adjusted to match the strain obtained from the saturated hysteresis loop of the HPTF experiment. Furthermore, as shown in the J2 model, the axial plastic strain and the axial stress are rather homogeneous throughout the gauge section, which justifies the application of the distortions of a volume element close to the surface as boundary conditions to the RVE, as the plastic strains observed in the RVE will be representative for the entire gauge section.

It is noted here, that the RVE is exposed to periodic boundary conditions to avoid free surfaces, which would severely influence the stress field within the RVE and produce finite size effects. This simplification is justified because we do not attempt to study crack initiation, for which free surface effects might play a crucial role.

In the first step to calibrate the CP parameters, the HPTF experiment with R=−1 and $\theta =7.5^{\circ }$ is used. Only the kinematic hardening parameters η and μ can be calibrated in this way, whereas the isotropic hardening parameters $\tau_{c}^{f}$,$h_{0}$, and $p_{2}$ are temporarily set to zero and will be calibrated in a second step as described below. To validate the correctness and generality of the CP parameters obtained by this procedure, the calibrated model is verified by comparing the simulation results with experiments for R=0, $\theta =10^{\circ }$ and R=0, $\theta =15^{\circ }$. Figure 10 shows that the stress–strain hysteresis loops obtained from HPTF simulations using the RVE model closely match the experiments for all loading cases, which confirms the predictive capability of the parameterized model.

The calibration of the isotropic hardening parameters ($\tau_{c}^{f}$,$h_{0}$, and $p_{2}$) and the remaining kinematic hardening parameter *m*, requires matching the evolution of maximum and minimum stress recorded during all cycles between simulation and experiment. Again, this is done for loading with ($R=-1,\;\theta =7.5^{\circ }$), and the result is validated by comparison with the other cases. As seen in Figure 11, a satisfying agreement between simulation and experiment could be achieved in this way. All resulting CP parameters are given in Table 5 and referred to as set 1.

As indicated in the parametric study shown in Appendix A, the axial creep behavior is also employed for the calibration of the CP parameters. Figure 12 shows the axial creep in the RVE during HPTF simulations. The results indicate that the CPFE model does not only predict the cyclic properties but also the axial creep behavior for the different HPTF loading cases very well.

In the next step, the CPFE model calibrated with HPTF experiments (set 1) is used to predict material behavior under uniaxial fatigue loading. It is pointed out here that it is expected that the material parameters fitted to more complex loading situations should be able to describe the material behavior under uniaxial loading (see, for example, the study of de Castro e Susa [36], where the authors tested various protocols to determine material parameters for cyclic plasticity using inverse procedures. However, the comparison of the simulation and experiments shown in Figure 13a exhibits the limitations of this approach, as the predicted and measured stress–strain hysteresis loops for uniaxial loading differ significantly from the parameters of set 1 that yield good comparability between experiment and simulation for the HPFT case, as seen in Figure 13b.

To understand the reason for this discrepancy in the prediction of the CPFE model, the experimental stress–strain hysteresis loops for HPTF and uniaxial fatigue tests are analyzed more closely. Figure 14 shows the comparison of the experimental results for saturated hysteresis loops under HPTF loading ($\theta =7.5^{\circ },\;R=-1$) and uniaxial fatigue loading ($E_{33}=0.55\%,\;R=-1$). Since the elastic stiffness in the former case is dominated by the materials’ shear modulus and in the latter case by Young’s modulus, the stress is normalized by the respective elastic quantities, which are estimated from the stiffness parameters given in Table 3 as Young’s modulus Y=185.2 GPa and shear modulus G=76.5 GPa.

It is seen that the effective slopes of the elastic branches of the HPTF and uniaxial hysteresis loops are significantly different, even after the proper scaling is applied, and that the elastic response of the material under uniaxial fatigue is approximately 1.46 times stiffer than under HPTF loading.

Due to this difference in the elastic slope, the effective elastic parameters required for the uniaxial fatigue simulations are different from those used for the HPTF simulation, which can be controlled by the scaling factor λ in this work. A similar change in the stress–strain relationship due to pre-straining has been reported in the experimental work of Vrh et al. [37]. A degradation of Young’s modulus by more than 20% during fatigue tests was reported for dual-phase steels in [38]. It can be interpreted from the experimental work of Wilshire and Willis [39], Stout et al. [40], and Schneider et al. [41] that the magnified effect of defects (especially dislocation structures) due to pre-straining in HPTF experiments can lead to the change in the stress–strain behavior. It is also pointed out here that the temperature increase during the HPTF experiments is significantly higher than that during the uniaxial tests (200 K vs. 50 K), which might also lead to a reduction in the elastic stiffness and the yield stress of the austenitic steel.

To demonstrate the possibility of calibrating the material parameters used for CPFE simulations to describe uniaxial fatigue behavior, the scaling parameter for elastic properties λ, the critical resolved shear stress $\tau_{0}$, and the isotropic hardening parameters ({$\tau_{c}^{f},h_{0}$}) are re-calibrated to match uniaxial experiments, while keeping the kinematic hardening parameters unchanged. The re-calibrated uniaxial parameters, referred to as set 2, are given in Table 6. The CPFE results obtained for both loading cases are plotted in Figure 13 as set 2. It is seen that this data set describes the stress–strain hysteresis under uniaxial loading very well, whereas the results for HPTF are not represented in an acceptable manner for set 2. A closer analysis of the material parameters of set 1 and set 2 reveals that the ratio of the fitted elastic constants in the CP models of both loading cases amounts to 1.51, which is close to the ratio of the effective stiffness values read from the experimental curves, which takes a value of 1.46. Furthermore, the scaling parameter λ used to adapt the elastic constants from DFT calculations to fit the uniaxial experiments is 0.95 and thus close to unity, which demonstrates the predictive capabilities of DFT methods concerning the calculation of elastic parameters. Generally, this finding that the transfer of crystal parameters obtained by inverse analysis of data referring to one experiment cannot be simply transferred to predict another experiment indicates a limitation in our current constitutive models and their parameters.

However, this analysis also reveals, that the kinematic hardening parameters obtained from uniaxial fatigue or HPTF loading are consistent, whereas the effective elastic parameters and, with them, some basic plastic properties, such as critical resolved shear stress and isotropic hardening parameters, take different values for the two different loading cases. The difference in these parameters can possibly be attributed to the higher temperatures that occur during HPTF testing or to different dislocation structures that result from multiaxial vs. uniaxial deformations. However, more research is required to draw a firm conclusion and to derive a model that describes these differences quantitatively. It is also pointed out here, that care needs to be taken when transferring material parameters obtained for particular loading cases to different scenarios.

## 5. Conclusions

In this work, we investigated the mechanical behavior of austenitic stainless steel 1.3808 (commercial name P558), exposed to high-pressure torsion fatigue (HPTF), i.e., overlaid constant axial and cyclic torsional loading. Experimental tests revealed that under such multiaxial reversed loading, the material undergoes plastic deformation in the axial direction if the amplitude of the cyclic torsion exceeds a critical value, even if the axial stress is well below the yield stress of the tested material. Since this time-dependent axial deformation occurs under a constant axial stress that is smaller than the yield stress, it can be considered a low temperature creep phenomenon, although its mechanisms are the same as those for plastic deformation and do not involve diffusive processes. Such material behavior has not yet been widely described in the literature. However, as multiaxial loading occurs frequently in technical systems or medical implants, it is essential to understand the mechanisms leading to this kind of plastic deformation. In order to accomplish this, two different numerical models, J2 and crystal plasticity (CP), were employed in finite element simulations of the HPTF experiments. The results indicate that the J2 model with an associated flow rule is not suitable to describe material behavior under such multi-axial cyclic loading cases. In contrast, the CP model with an Ohno–Wang kinematic hardening model reveals a good agreement with the experiment and even was able to predict the material response for different HPTF loading cases after fitting the material parameters to experimental data from one individual test.

A comparison between uniaxial fatigue experiments and HPTF loading demonstrated that there was a significant difference in the elastic branches of the respective stress–strain hysteresis loops. The origin of this discrepancy might lie either in the significantly higher temperature that builds up in the material during HPTF testing, in different dislocation structures that develop during the different test conditions, or in a combination of both. Consequently, the CP model calibrated by an inverse procedure based on HPTF data is incapable of matching the results of uniaxial fatigue experiments. However, it was demonstrated that a re-calibration of the material stiffness tensor and isotropic hardening parameters by a scalar factor, while leaving the kinematic hardening parameters unchanged, leads to a good description of the data obtained from uniaxial fatigue experiments. To understand the correlation between the constitutive parameters and the resulting material behavior in more detail, a parametric study of the CP kinematic hardening model under HPTF loading was performed, and the resulting stress–strain hysteresis loops and axial creep deformation were analyzed. The most important result of this case study is that a reliable parameter identification by an inverse modeling approach based on experimental data is possible, and that including axial deformation into the parameter fitting procedure improves the calibration of the parameters for kinematic hardening. Generally, our results indicate that material parameters that are identified by an inverse analysis of experimental data from more complex situations can be used to describe simpler loading conditions, whereas material parameters identified from simpler tests typically cannot be used to describe more complex loading situations. However, it must also be pointed out here that there are cases where material parameters obtained by inverse analysis of some test conditions simply cannot be generalized to other situations, which is a clear limitation of current constitutive models and the parameters they rely on.

The finite element model used for the simulation of the material response based on crystal plasticity employed a simple representative volume element (RVE) that mimics the crystallographic texture obtained from the EBSD analysis of the P558 steel specimens used in the experiment. This RVE contained 512 cubic grains, thus ignoring microstructural features like grain shape or annealing twins—a simplification that was necessary to minimize the numerical effort in this work. In forthcoming studies, it is planned to investigate the influence of grain shapes, annealing twins, and crystallographic texture on the material behavior under HPTF loading in more detail by considering more realistic grain geometries in the RVE simulations.

## Figures and Tables

**Figure 1 materials-16-01367-f001:**
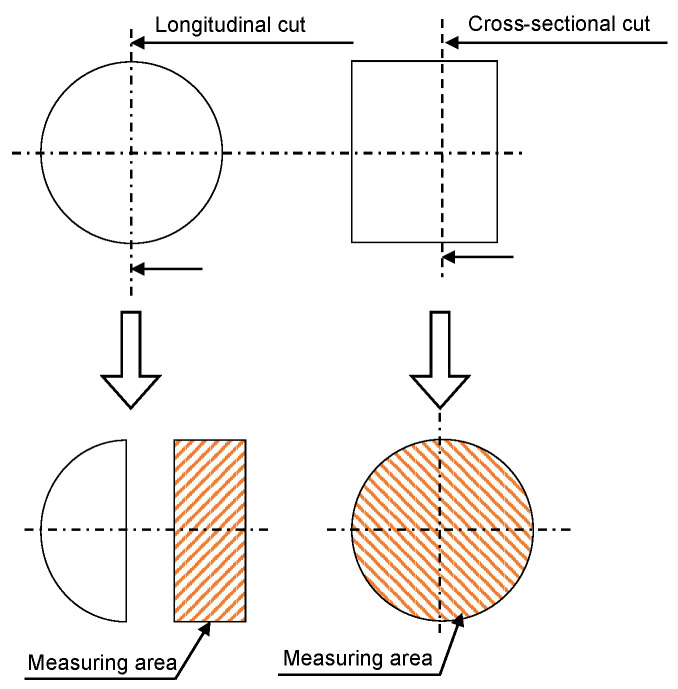
Schematic view of the cutting directions of the specimens for EBSD Analysis.

**Figure 2 materials-16-01367-f002:**
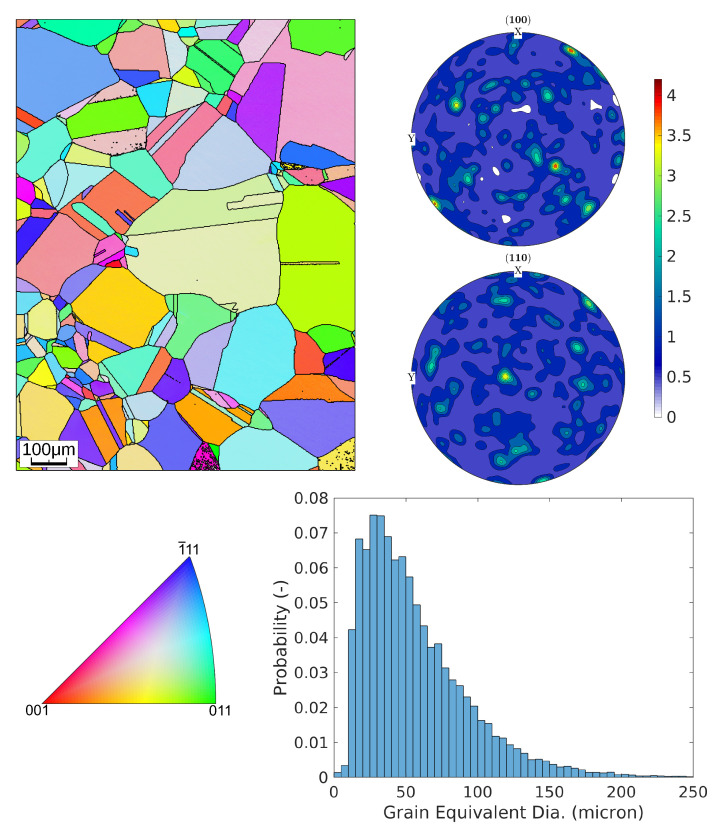
(**top left**) IPF-Z plot of one of the EBSD data sets, (**bottom left**) corresponding IPF key, (**top right**) pole figure contour plot of the ODF estimated by combining all EBSD data sets (i.e., ≈19,000 grains), (**bottom right**) the grain equivalent diameter histogram.

**Figure 3 materials-16-01367-f003:**
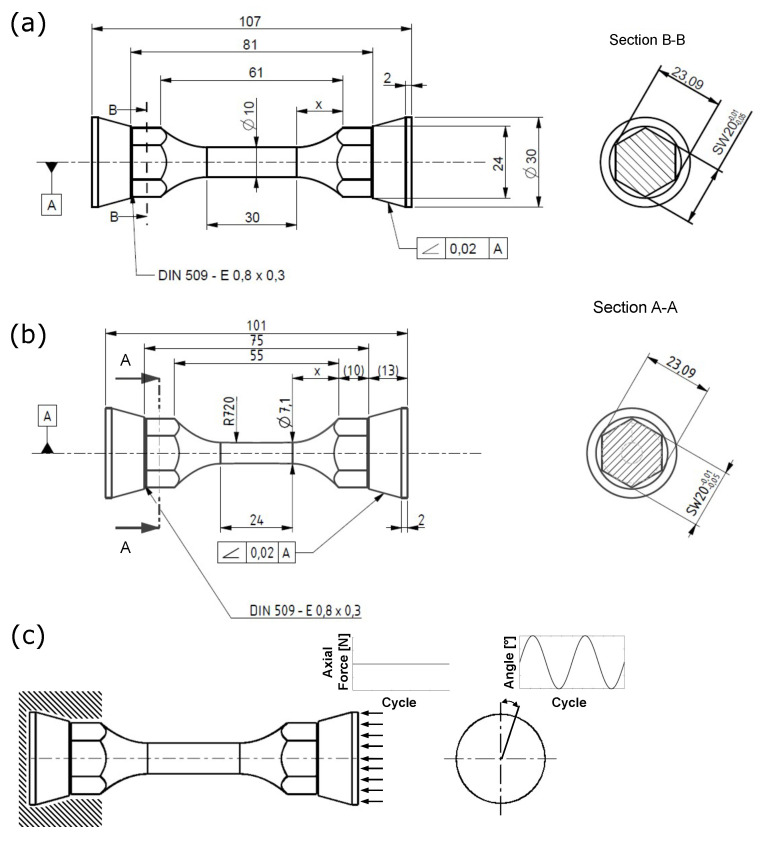
Figure showing the dimensions and surface finish for (**a**) HPTF specimen, (**b**) uniaxial fatigue specimen, and (**c**) HPTF loading of the specimen, indicating the constant axial force and the cyclic torsional angles.

**Figure 4 materials-16-01367-f004:**
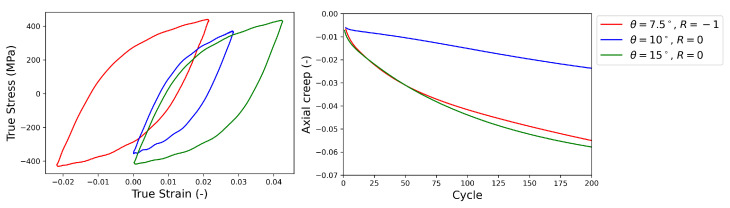
(**left**) Experimental stress–strain hysteresis loops recorded for uniaxial fatigue after the saturation of the mean stress and (**right**) axial creep for the HPTF loading cases given in the legend, where θ is the amplitude of the cyclic torsion loading, and *R* is the ratio of the minimum vs. maximum torsion angle.

**Figure 5 materials-16-01367-f005:**
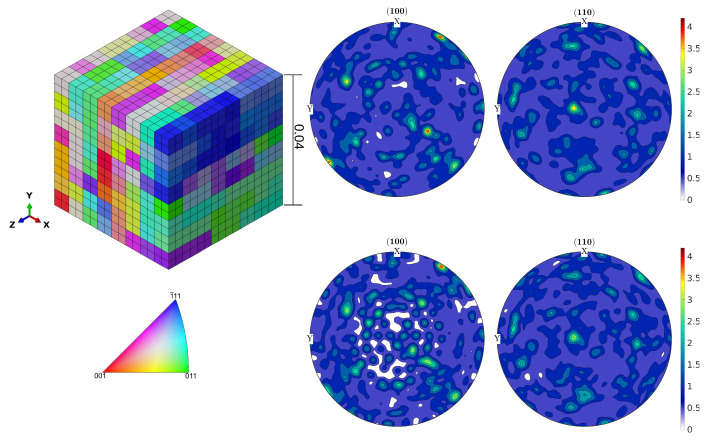
(**top left**) Graphical representation of the used RVE with 512 cubic grains and 8 finite elements per grain, (**bottom left**) grains are colored according to their orientation and the corresponding IPF key. Contour plots of the ODF pole figures from the orientation set obtained by combining all the EBSD data sets of about 19,000 grains (**top right**) and 512 discrete orientations in the RVE (**bottom right**).

**Figure 6 materials-16-01367-f006:**
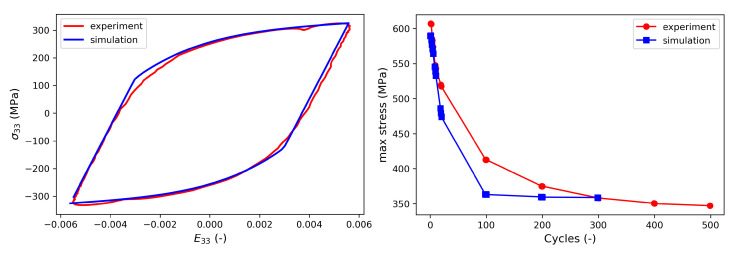
(**left**) Stress–Strain hysteresis loop and (**right**) reduction of the maximum stress in uniaxial fatigue tests with a strain amplitude of $E_{33}=0.55\%$ and an amplitude ratio of R=−1.

**Figure 7 materials-16-01367-f007:**
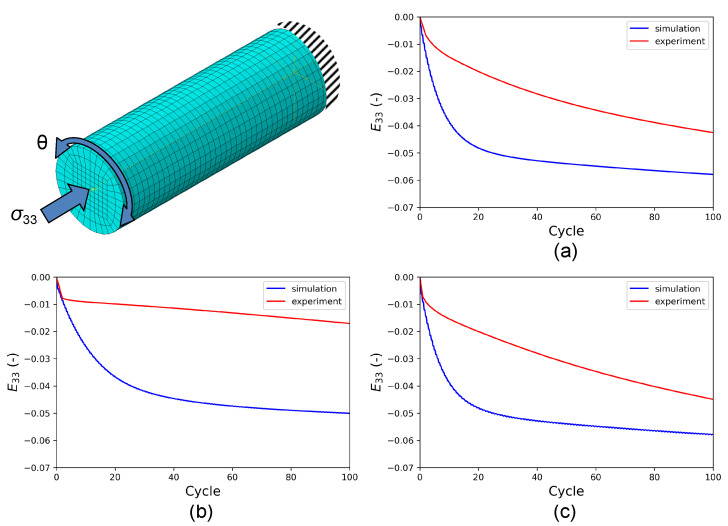
(**top left**) FE model used to simulate HTPF loading with a J2 plasticity model and resulting axial creep strain ($E_{33}$) of simulation and experiment vs. cycle number. The axial pressure amounted to $\sigma_{33}=250$ MPa and three different combinations of amplitude ratios *R* and cyclic torsion amplitudes θ are shown: (**a**) $R=-1,\;\theta =7.5^{\circ }$, (**b**) $R=0,\;\theta =10^{\circ }$, and (**c**) $R=0,\;\theta =15^{\circ }$.

**Figure 8 materials-16-01367-f008:**
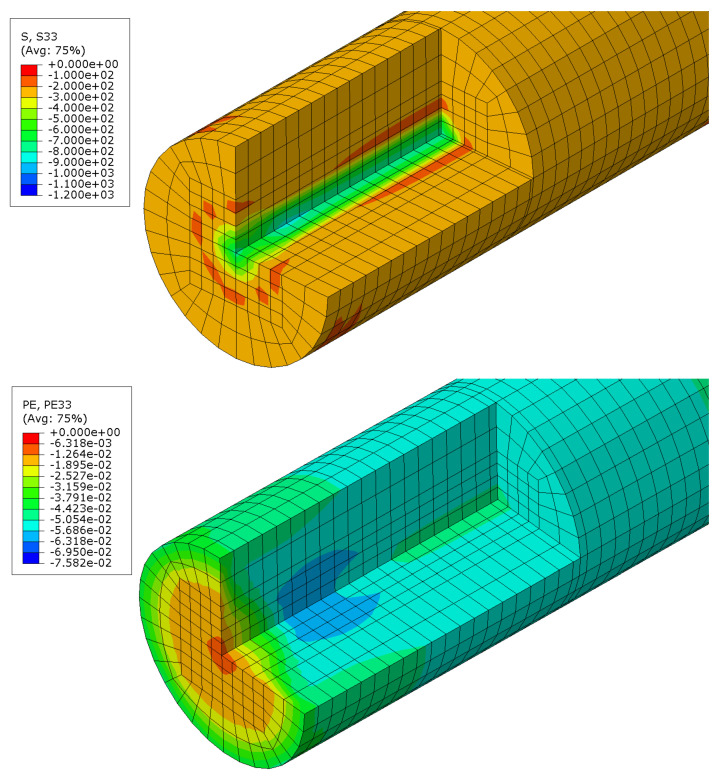
Contour plot of the axial stress in MPa (**top**) and the axial plastic strain (**bottom**) along a cut through the J2 model.

**Figure 9 materials-16-01367-f009:**
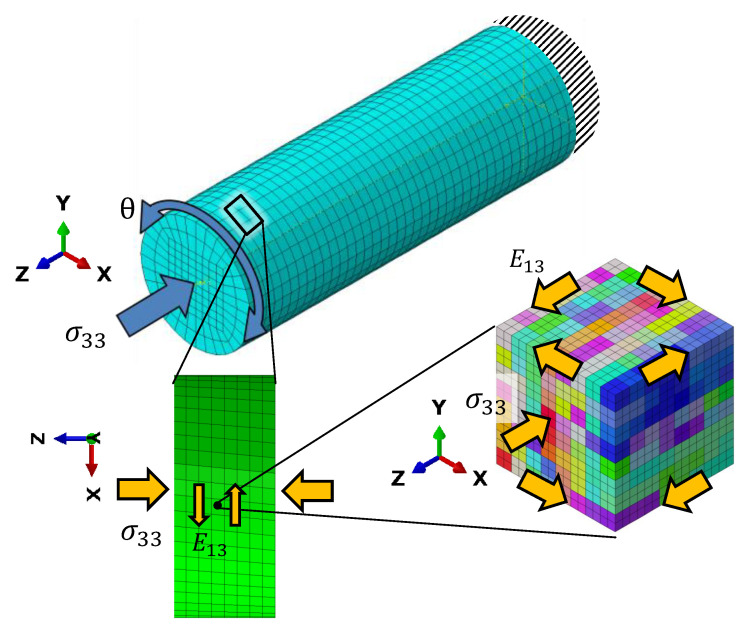
To mimic HPTF loading conditions on the length scale of the RVE used for CPFEM simulations, the distortions of a small volume element close to the surface of the cylindrical FE model, representing the gauge section of the HPTF specimens, are applied as boundary conditions to the RVE.

**Figure 10 materials-16-01367-f010:**
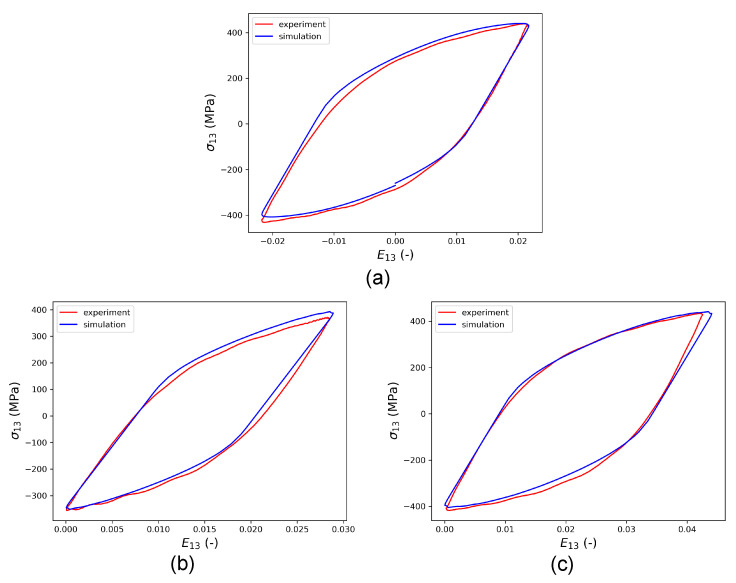
Comparison of saturated HPTF stress–strain hysteresis loops between simulation and experiment; the constant axial pressure amounts to $\sigma_{33}=250$ MPa in all cases and cyclic shear loading $E_{13}$ is applied to mimic the cyclic torsion in the experiment. (**a**) The result of calibration for HPTF loading with $R=-1,\;\theta =7.5^{\circ },\;E_{13}=0.02$. The two subfigures in the bottom represent predictions of the model for (**b**) $R=0,\;\theta =10^{\circ },\;E_{13}=0.028$, and (**c**) $R=0,\;\theta =15^{\circ },\;E_{13}=0.04$.

**Figure 11 materials-16-01367-f011:**
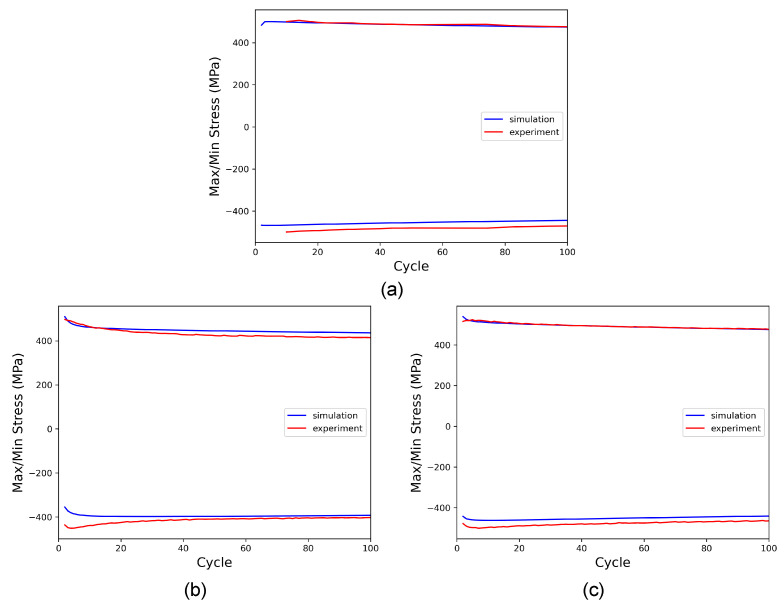
Comparison of maximum and minimum stress obtained in experiment and simulation during torsional cycles with (**a**) $R=-1,\;\theta =7.5^{\circ }$, comparison of cyclic softening behavior for the loading cases (**b**) $R=0,\;\theta =10^{\circ }$, and (**c**) $R=0,\;\theta =15^{\circ }$.

**Figure 12 materials-16-01367-f012:**
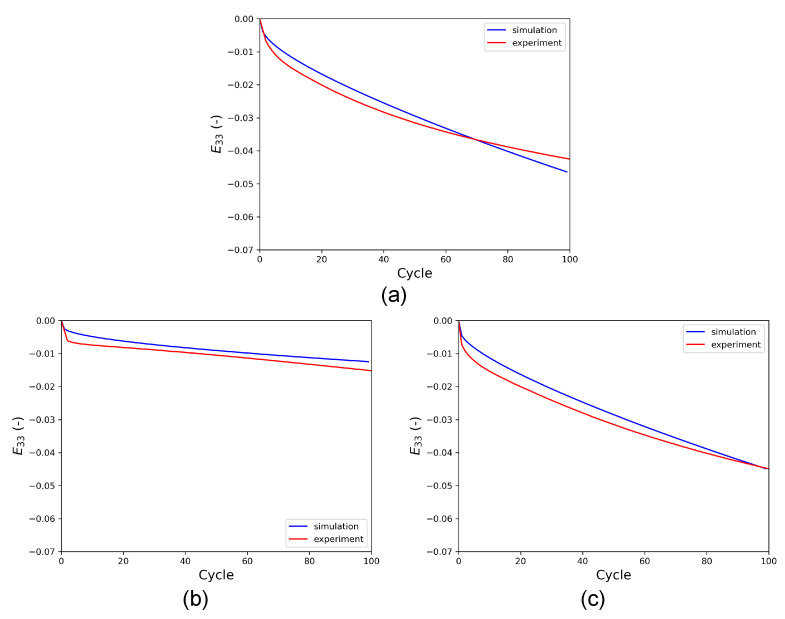
Axial creep strain ($E_{33}$) vs. the number of cycles obtained from HPTF simulations with the CPFE model and from experiment for a constant axial pressure of $\sigma_{33}=250$ MPa and cyclic torsional loading: (**a**) $R=-1,\;\theta =7.5^{\circ }$) used for calibration, (**b**) $R=0,\;\theta =10^{\circ }$, and (**c**) $R=0,\;\theta =15^{\circ }$. Cases (**b**,**c**) represent predictions of the model.

**Figure 13 materials-16-01367-f013:**
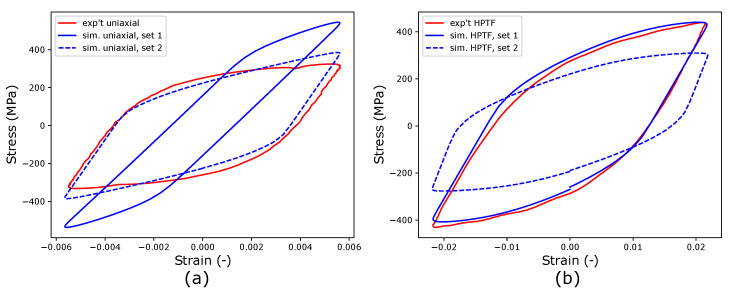
Comparison of experimental and simulated stress–strain hysteresis loops (**a**) for uniaxial fatigue with the strain amplitude $E_{33}=0.55\%$ and (**b**) under HPTF loading with (R=−1, $\theta =7.5^{\circ }$, $E_{13}=0.02$). The material parameters for set 1 (Table 5) were obtained using an inverse analysis of the HPTF experiments, whereas the parameters of set 2 (Table 6) were fitted to uniaxial experiments.

**Figure 14 materials-16-01367-f014:**
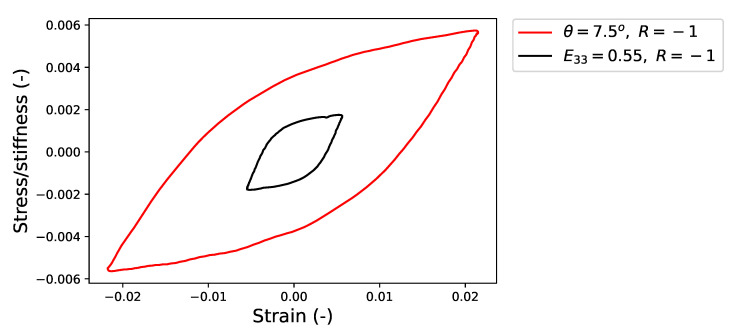
Comparison of the hysteresis loops in the regime where the stress amplitude is constant for the uniaxial fatigue experiment ($E_{33}=0.55\%,\;R=-1$) and the HPTF loading case ($\theta =7.5^{\circ },\;R=-1$), where θ and $E_{33}$ are the amplitudes of the cyclic deformation. The values of all stresses in the plot are normalized with the corresponding stiffness parameters, i.e., Young’s Modulus Y=185.2 GPa for uniaxial fatigue and shear modulus G=76.506 GPa for the torsional loading during HPTF.

**Table 1 materials-16-01367-t001:** Chemical composition of P558.

C	Si	Mn	P	S	Cr	Ni	Mo	Cu	V	Nb	N
0.19	0.41	12.20	0.01	0.00	17.15	0.02	2.89	0.02	0.01	0.03	0.54

**Table 2 materials-16-01367-t002:** Details of uniaxial fatigue and high pressure torsion fatigue (HPTF) experiments. Axial stress and strain components are denoted by indices “33”.

Type	*R*	Amplitude	Frequency	Axial Pressure
Uniaxial	−1	$E_{33}=0.55\%$	2.5 Hz	-
HPTF	−1	$\theta =7.5^{\circ }$	2.5 Hz	$\sigma_{33}=250$ MPa
HPTF	0	$\theta =10^{\circ }$	2.5 Hz	$\sigma_{33}=250$ MPa
HPTF	0	$\theta =15^{\circ }$	2.5 Hz	$\sigma_{33}=250$ MPa

**Table 3 materials-16-01367-t003:** Ideal elastic stiffness parameters $C_{11}$, $C_{12}$, and $C_{44}$ obtained by DFT calculations, resulting homogenized Young’s modulus *Y*, and scaling factor λ.

$C_{11}$	$C_{12}$	$C_{44}$	*Y*	λ
(GPa)	(GPa)	(GPa)	(GPa)	(-)
263.159	122.644	76.506	185.184	0.63

**Table 4 materials-16-01367-t004:** Calibrated J2 model parameters.

$C_{1}$	$C_{2}$	$g_{1}$	$g_{2}$	$\sigma^{m}$	*Q*	*b*
(MPa)	(MPa)	(-)	(-)	(MPa)	(MPa)	(-)
43,106.44	4192.81	513.95	0.0	545.82	−300	3.17

**Table 5 materials-16-01367-t005:** Calibrated CPFE model parameters (set 1).

λ	${\dot{\gamma }}_{0}$	$p_{1}$	$\tau_{0}$	$\tau_{c}^{f}$	$h_{0}$	$p_{2}$	η	μ	*m*
(-)	(1/s)	(-)	(MPa)	(MPa)	(MPa)	(-)	(MPa)	(-)	(-)
0.63	0.001	25	235	184.2	−25.0	2	16,888.0	125.0	2.0

**Table 6 materials-16-01367-t006:** Re-calibrated CPFE model parameters for uniaxial fatigue loading (set 2). The remaining material parameters are unchanged with respect to set 1 and given Table 5.

λ	$\tau_{0}$	$\tau_{c}^{f}$	$h_{0}$
(-)	(MPa)	(MPa)	(MPa)
0.95	188	92	−130.0

## Data Availability

Data will be made available upon request.

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
