# Peer review of "Mechanical Behavior of Austenitic Steel under Multi-Axial Cyclic Loading"

_materials, 2023, doi:10.3390/ma16041367_

Round 1

Reviewer 1 Report

Minor revision as listed below is required.

-       the 82 tensile strength of the material was 937 MPa and the yield strength was 595 MPa.

Need to explain how above-mentioned values are obtained, and testing conditions, if available.

-       It appears that EBSD is not defined.

-       Dimensions of the as-received material is missing. In Figure 1, dimensions are not provided. It is unclear what the initial conditions of the as-received material is.

-       The material was solution annealed at 1150â—¦C for 45 min,

What is the solution applied here?

-       bar material were cut both in longitudinal and cross- 84 sectional directions according to Figure 1.

The original data measured for grain orientation function of both directions should be provided (maybe in supplementary? So we can see directly if the sample is isotropic after annealing).

-       limiting the temperature increase of 129 samples

Pls provide the actual room temperature, and typical temperature vs. time relationship during testing, if possible.

-       the temperature increase may reach values over 200 K

is it possible to test the same samples at room temperature +200 K for comparison?

-       Eqn 4 is based on a linear relationship between stress and strain. It may not be applicable in the cases as we have here. It is necessary to point it out that this is an approximation.

-       and frequency of 2.5 Hz, combined with a monotonous 134 constant compressive stress 250 or 350 MPa.

However, in Table 2, only 250 MPa is mentioned. Why is 0 MPa pressure not included as well?

Author Response

We sincerely thank the reviewer for the detailed comments and suggestions. Below we address each point raised by the reviewer and print our answer in red ink:

  • Line 82: the tensile strength of the material was 937 MPa and the yield strength was 595 MPa. Need to explain how above-mentioned values are obtained, and testing conditions, if available.

These values have been obtained as mean values of three standard tensile tests.

  • It appears that EBSD is not defined.

This abbreviation has now been declared upon its first use.

  • Dimensions of the as-received material is missing. In Figure 1, dimensions are not provided. It is unclear what the initial conditions of the as-received material is.

The material is delivered in form 35mm x 1000mm cylindrical bars, received in unknown heat treatment state. Those bars are cut in 120mm pieces and heat threated as explained, where material for HPTF and tensile test samples did undergo the same process. Afterwards, the samples are manufactured in the specified geometries.

  • The material was solution annealed at 1150°C for 45 min, What is the solution applied here?

The term “solution annealed” means that the material was heat treated to generate a homogeneous distribution (“solution”) of all alloying elements and to remove any cold working from manufacturing process. This has been clarified in the text now.

  • Line 84: bar material were cut both in longitudinal and cross-sectional directions according to Figure 1.

The original data measured for grain orientation function of both directions should be provided (maybe in supplementary? So we can see directly if the sample is isotropic after annealing).

The microstructure has been analyzed along both cuts but since the grains are rather equiaxed only one micrograph is shown here. For the characterization of the crystallographic texture, the orientation distribution functions (ODF) of both cuts have been combined, considering the 90° rotation between them.

  • Line 129: limiting the temperature increase of samples

Pls provide the actual room temperature, and typical temperature vs. time relationship during testing, if possible.

The actual room temperature was between 20 ± 2 °C with 40-60% relative humidity with climatized laboratory conditions. This information has been added to the text.

  • the temperature increase may reach values over 200 K

is it possible to test the same samples at room temperature +200 K for comparison?

Since the samples heat up due to the cyclic deformation, a precise control of the temperature to obtain reference experiments is close to impossible.

- Eqn 4 is based on a linear relationship between stress and strain. It may not be applicable in the cases as we have here. It is necessary to point it out that this is an approximation.

This point has been clarified in the text.

- line 134: and frequency of 2.5 Hz, combined with a monotonous constant compressive stress 250 or 350 MPa.

However, in Table 2, only 250 MPa is mentioned. Why is 0 MPa pressure not included as well?

In uniaxial fatigue testing, the axial pressure results from the material response and cannot be given in this table.

Reviewer 2 Report

1. In figure 7, what makes the big difference between simulation and experiment?

2. It is better to explain the author's future work in section "Conclusions"

Author Response

We sincerely thank the reviewer for the comments. Please find our replies below in red ink:

  1. In figure 7, what makes the big difference between simulation and experiment?

As described in the text, the simple J2 plasticity model with an associated flow rule is not able to capture the deformation behavior under the multiaxial load with a sufficient accuracy.

  1. It is better to explain the author's future work in section "Conclusions"

The outlook at the end of the Conclusions is explained in more detail.

Reviewer 3 Report

This paper investigates the mechanical behavior of low nickel austenitic steel under multi-axial cyclic load. The experimental work is good and the explanations for the observations are reasonable. References are appropriate. It is suggested to publish in Materials.

Some comments are listed as following.

1. In general, the novelty of the work is slightly inadequate. The present paper need to highlight the most significant finding of the manuscript.

2. The introduction section and the rational of this work is coarse and not well explained, please it is necessary to explain in detail which is bringing new this work with respect to other studies. Please explain in detail which is the originality of this work.

3. The conclusions are formulated very general without summarizing what was important in the work.

Author Response

We sincerely thank the reviewer for the comments. The novelty of this work has been highlighted in the introduction and the conclusions.

Reviewer 4 Report

Dear Authors,

Your paper reflects the interesting problem for scientists and engineers in the experimental and numerical approaches. Nevertheless, some corrections are required. Please follows all comments and suggestions.

Reviewer

Author Response

We sincerely thank the reviewer for the careful reading of our manuscript and the detailed suggestions. The recommended amendments have been put into effect where it seemed adequate and the two suggested references have been added.

Please note that for Fig. 4 and throughout our work we actually reported a creep strains for a constant axial pressure.

Round 2

Reviewer 4 Report

Dear Authors,

Your paper is prepared in the correct manner and it very well reflects the problem considered.

Reviewer